# Effects of Air Pollution Control on Urban Development Quality in Chinese Cities Based on Spatial Durbin Model

**DOI:** 10.3390/ijerph15122822

**Published:** 2018-12-11

**Authors:** Yanchao Feng, Xiaohong Wang, Wenchao Du, Jun Liu

**Affiliations:** School of Economics and Management, Harbin Institute of Technology, Harbin 150001, China; m15002182995@163.com (Y.F.); 17B910027@stu.hit.edu.cn (W.D.); liujun6098@163.com (J.L.)

**Keywords:** air pollution control, urban development quality, spatial Durbin model

## Abstract

With the rapid development of urbanization, industrialization, and motorization, a large number of Chinese cities have been affected by heavy air pollution. In order to promote the development quality of Chinese cities, mixed regulations to control air pollution have been implemented under the lead of government. The principal component analysis and efficacy coefficient method are used to estimate urban development quality, according to the panel data of 285 prefecture-level cities in China over the period 2003–2016. On this basis, the paper uses the spatial Durbin model to study the direct impact and the spatial spillover effect of air pollution control on urban development quality in China. Results show that the control of smoke and dust has improved urban development quality in China, however, the control of sulfur dioxide has led to the decline of urban development quality in China. Furthermore, the impact of air pollution control on urban development quality in the eastern region is of great significance in statistical tests, while the situation in the central and western regions has not passed the test, implying the spatial heterogeneity among different regions. The different effects of air pollution control on urban development quality in different regions also illustrate the consciousness and supervision of local governments’ environment protection. Finally, the effects decomposition of the influencing factors based on spatial Durbin model (SDM) also supports the robust findings. Promoting the upgrading of energy consumption structure, raising awareness of environmental protection and supervision, and strengthening cooperation of different regions are suggested. Further recommendations are provided to improve the conceptual design and increase the credibility of our research. Our study not only provides new evidence on the impact of air pollution control on urban development quality in China, but also proposes a new perspective to promote urban development quality in China.

## 1. Introduction

Over the past decades, the production of energy from various sources (e.g., coal, crude oil, natural gas) has increased rapidly in China, partly caused by the relatively comfortable and prosperous life sought by people, however, along with prosperity, a large number of Chinese cities are affected by heavy air pollution [1]. The unprecedented consequences not only pose challenges to the provision of jobs, housing, and infrastructure, but also exert more pressure on urban land management, spatial equity, and more generally sustainable development [2]. In order to promote the sustainable development of economy and society, the Chinese government has made some active efforts to reduce pollutant emissions and protect the environment. However, despite the government’s environmental protection efforts, China’s environmental quality seems to be continuously deteriorating, and the amount of the emissions of main air pollutants remains persistently high [3]. In recent years, the relationship between air pollution control and economic growth has attracted the attention of many scholars.

With the upgrading of developmental concepts, urban development quality has been transformed from the initial pursuit of urbanization rate or rapid economic growth to the pursuit of sustainable development of economy, society, and environment [4,5,6,7]. Characterized by the rise of central business districts (CBDs) for advanced business services, newly formed production centers in suburban areas, and spatially segregated neighborhoods dividing the rich and the poor, the unique patterns of urban development in China has occurred at an unprecedented rate [8,9,10,11,12,13]. However, the massive construction boom across the whole nation in recent years has left abundant housing, factories, commercial facilities, and new urban districts with incredibly and inexcusably low occupancy rates, known as ‘ghost cities/towns’ [14]. Therefore, ignoring social welfare and environmental improvement, the immense increase of urban built-up areas or rapid economic growth can not reflect urban development quality comprehensively [15]. Given the fact that enormous regional disparities in levels of development and urbanization across China, it is very necessary to use nationally representative data to further study urban development quality [16].

Nevertheless, there are few studies that explore the global characteristics and the driving factors of urban development quality in China, especially the impact of air pollution control on urban development quality has not been comprehensively and thoroughly examined. Moreover, existing literature overemphasizes the importance of air pollution control on the economic dimension of urban development quality, while the subject of the social and the environmental dimension of urban development quality is particularly under-researched [17,18]. Therefore, more sophisticated studies are needed to examine the impact of air pollution control on urban development quality, which is of great and practical significance for guiding the adjustment and optimization of existing air pollution control policies in different regions. Most researchers, academics, and practitioners took it for granted that air pollution control would be effective, and air pollution control would naturally lead to the reduction of environmental pollution and the improvement of environmental quality [19,20,21,22]. However, this perception might be misleading. For different objects, samples and methods, the cost and effect of air pollution control are not always consistent [23,24,25,26]. Moreover, there is no literature on exploring the impact of air pollution control on the quality of regional urban development. Furthermore, the level of urban development quality and the degree of air pollution control vary in the eastern, central, and western China [16]. Hence, it is necessary to investigate the impact of air pollution control on urban development quality across different regions in China.

As mentioned above, previous related literature is insufficient. To analyze the effects of air pollution control on urban development quality in different regions and reveal the spatial effect of air pollution control on regional urban development quality, this paper has made following contributions to knowledge. Firstly, the effective mechanisms of diversified air pollution control are different. Based on the difference between pre-treatment and post-treatment strategies, air pollution control is classified into two types such as energy saving air pollution control and emission reduction air pollution control. Based on the differences of industrial air pollutants, the energy saving type is measured by the emission of sulfur dioxide per GDP and the smoke and dust per GDP; correspondingly, the emission reduction type is measured by the removal rate of sulfur dioxide and smoke and dust. Secondly, environmental problems are characterized as different regions due to the imbalance of urban development quality. Therefore, in this paper, China is divided into three classical regions: the eastern region, the central region and the western region, to explore the regional difference of the impact of air pollution control on urban development quality. Lastly, the spatial Durbin model is introduced to examine the spatial dependence of air pollution control on urban development quality in China, which is conductive to revising policies of environmental protection for regional government.

## 2. Methodology

Tobler’s first law of geography indicates that all phenomena in space are linked, but the connection intensity is stronger at near distances versus long distances [27]. This law well represents impacts of air pollution control on urban development quality because environmental pollution has strong trans-regional and agglomerate character. In other words, air pollution control and urban development quality show significant spatial dependence. Consequently, we select the spatial Durbin model (which allows testing the existence of both endogenous and exogenous interactions) to investigate the impacts of air pollution control on urban development quality in Chinese cities [28]. Actually, the spatial regression models are frequently applied to economic, environmental, and ecological modeling [29].

### 2.1. Spatial Durbin Model

This study aims to investigate the direct and spillover effects of air pollution control on urban development quality in Chinese cities. Spatial Durbin model (SDM) can examine the influence of the dependent variable affected by the local area variables, as well as the dependent and independent variables in neighboring areas, which is a general form of the spatial lag model (SLM) and spatial error model (SEM). In this way, the spatial Durbin model (SDM) is more suitable for the objectives of this study [30,31,32]. Its basic form is: (1)$$Y_{it}=\rho \sum_{j=1}^{N}w_{ij}Y_{jt}+\beta X_{it}+\theta \sum_{j=1}^{N}w_{ij}X_{jt}+\mu +\varepsilon_{it}$$
where *Y_it_* is the dependent variable in city *i* at year *t*; *w_ij_* corresponds to the spatial connectivity assigned to city *j* by city i(j≠i); *ρ* is the spatial parameter of interest, which reflects the endogenous spatial interaction between city *i* and its neighboring cities; *β* is a vector of the coefficients of the explanatory variables; *X_it_* is the explanatory variables of city *i* that explains its urban development quality; θ reflects exogenous interaction effects, which creates an average of explanatory variable values from neighboring cities which are added to the set of conventional explanatory variables; the matrix $w_{ij}X_{jt}$ denotes the spatial lag effects associated with explanatory variables; μ denotes the random-effects or the time and city fixed effects; and $\varepsilon_{it}$ represents an error term uncorrelated with the explanatory variables across cities and over time, which is assumed to be normally distributed.

Based on Equation (1), the spatial Durbin model (SDM) of the impact of air pollution control on urban development quality is as follows: (2)$$\begin{array}{l} \ln UDQ_{it}=\rho \sum_{j=1}^{N}w_{ij}\ln UDQ_{it}+\beta_{1}\ln ES_{1,it}+\beta_{2}\ln ES_{2,it}+\beta_{3}\ln ER_{1,it}+\beta_{4}\ln ER_{2,it}\\ +\beta_{5}\ln LF_{it}+\beta_{6}\ln FD_{it}+\beta_{7}\ln HC_{it}+\beta_{8}\ln FDI_{it}+\beta_{9}\ln IU_{it}+\theta_{1}\sum_{j=1}^{N}w_{ij}\ln ES_{1,it}\\ +\theta_{2}\sum_{j=1}^{N}w_{ij}\ln ES_{2,it}+\theta_{3}\sum_{j=1}^{N}w_{ij}\ln ER_{1,it}+\theta_{4}\sum_{j=1}^{N}w_{ij}\ln ER_{2,it}+\theta_{5}\sum_{j=1}^{N}w_{ij}\ln LF_{it}\\ +\theta_{6}\sum_{j=1}^{N}w_{ij}\ln FD_{it}+\theta_{7}\sum_{j=1}^{N}w_{ij}\ln HC_{it}+\theta_{8}\sum_{j=1}^{N}w_{ij}\ln FDI_{it}+\theta_{9}\sum_{j=1}^{N}w_{ij}\ln IU_{it}\\ +\mu +\varepsilon_{it} \end{array}$$
where *UDQ* indicates urban development quality; *ES*_1_, *ES*_2_, *ES*_3_, *ES*_4_ indicate the emission intensity of sulfur dioxide, the emission intensity of smoke and dust, the removal rate of sulfur dioxide and the removal rate of smoke and dust respectively; *LF* indicates the shares of land leasing revenue in GDP; *FD* indicates the shares of both deposits and loans in GDP; *HC* indicates the number of college students per 10,000 people; *FDI* indicates the shares of foreign direct investment in GDP; and *IU* indicates the shares of the value of the tertiary industries in the value of the secondary industries.

To avoid bias caused by the coefficient estimate of the explanatory variable, the total effect should be divided into a direct and an indirect effect by using the partial derivative method [30,31], and the SDM can be transferred as follows: (3)$$Y_{t}=[{\left(I-\rho w\right)}^{-1}(\beta X_{t}+\theta wX_{t})]+{\left(I-\rho w\right)}^{-1}\varepsilon_{t}$$

The partial differential equation matrix for the *k* explanatory variable is as follows:(4)$${\left[\frac{\partial Y}{\partial X_{1k}},\frac{\partial Y}{\partial X_{2k}},\cdots \frac{\partial Y}{\partial X_{Nk}}\right]}_{t}={\left(I-\rho w\right)}^{-1}\left[\begin{array}{cccc} \beta_{k} & w_{12}\theta_{k} & \cdots & w_{1N}\theta_{k}\\ w_{21}\theta_{k} & \beta_{k} & \cdots & w_{2N}\theta_{k}\\ \vdots & \vdots & \ddots & \vdots \\ w_{N1}\theta_{k} & w_{N1}\theta_{k} & \cdots & \beta_{k} \end{array}\right]$$
where the average value of the diagonal elements represents the direct effect, the average value of the non-diagonal elements represents the indirect effect (spatial spillover effect), and the sum of the direct effect and the indirect effect is the total effect. Due to the existence of spatial dependence and time inertia, there will be spatial feedback effects on urban development between a certain city and the surrounding cities, resulting in a certain deviation between the decomposition results and the regression results of the spatial Durbin model (SDM) [28,29,30,31].

### 2.2. Spatial Weight Matrix

Spatial weight matrix ($w_{ij}$) is the core element of spatial panel data models. Two types of spatial weight matrix (the squared term of inverse distance matrix ($w_{ij}^{1}$) and the squared term of inverse distance and economic-based matrix ($w_{ij}^{2}$)) are adopted in this study to reflect two different spatial relations, which is defined as follows.
(5)$$w_{ij}^{1}=\{\begin{array}{c} 0,i=j\\ \frac{1}{{\left(d_{ij}\right)}^{2}},i\neq j \end{array}$$
(6)$$\begin{array}{l} g=diag(\bar{GDP_{1}}/\bar{GDP},\bar{GDP_{2}}/\bar{GDP},\cdots \bar{GDP_{N}}/\bar{GDP})\\ w_{ij}^{2}=\{\begin{array}{c} 0,i=j\\ \frac{g}{{\left(d_{ij}\right)}^{2}},i\neq j \end{array} \end{array}$$
where *d_ij_* is the greater-circle distance calculated based on the longitude and latitude between city i and city *j*, and $w_{ij}^{1}$ takes into account relations of all cities, which allows for testing all-with-all interactions in the whole territory; $\bar{GDP_{i}}$ and $\bar{GDP}$ present, respectively, the average GDP of city *i* and all cities during the study periods, and $w_{ij}^{2}$, which is set as all Chinese cities, are inter-connected in terms of both geographical and economic factors. We normalized the two spatial weight matrix to have row-sums of one and main diagonal elements of zero.

## 3. Data and Variables

According to the administrative level, Chinese cities can be divided into prefecture-level cities and county-level cities. Under the administrative division system, a prefecture-level city contains municipal districts and other units, such as county-level cities, counties, and towns. As for the concept of a city, a prefecture-level city in China often refers to the municipal districts, which resembles Western cities [33]. However, a county-level city does not have a clear central urban area and usually contains a large proportion of non-urbanized areas. Therefore, we choose prefecture-level cities as samples for this study. Due to data unavailability (some regions including Taiwan, Hong Kong, and Macau are excluded temporally due to unavailability, some cities have experienced administrative division adjustments in the past decade, and other cities have data missing for certain years), a panel data set on 285 prefecture-level cities (see Appendix A) over the period of 2003–2016 has been used (Figure 1).

### 3.1. Dependent Variable

To measure urban development quality comprehensively and accurately, 15 indexes relevant to the three dimensions (economic, social and environmental) should be considered as much as possible within the range of data availability (see Appendix B). To eliminate the effect of dimension and magnitude factors, 15 indexes of urban development quality in each year are normalized as follows: (7)$$y_{ji}^{*}=\frac{y_{ji}-\mu_{ji}}{\sqrt{\sigma_{ji}}}$$
where $y_{ji}^{*}$ is the normalized value of the related statistical factor j in the city i, between 0 and 1; $y_{ji}$ is the original value of the related statistic factor j in the city i of the 285 prefecture-level cities in each year; $\mu_{ji}$ is the average value of the related statistic factor j in the city i of the 285 prefecture-level cities in each year; and $\sigma_{ji}$ is the variance value of the related statistic factor j in the city i of the 285 prefecture-level cities in each year. After normalizing the original data with this equation, the normalized data variance is 1, and the average value is 0.

The most commonly used multivariate statistical analysis method is Principal Component Analysis (PCA), which can select several important variables to reduce the number of factors by using linear transformation [34]. Application of the varimax rotation of the normalized component loading allows us to obtain a clear system by maximizing component load differences and eliminating invalid components [35].

The model, with the sample set, is as follows: (8)$$Y={\left[\begin{array}{cccc} y_{11} & y_{12} & \cdots & y_{1p}\\ y_{21} & y_{22} & \cdots & y_{2p}\\ \vdots & \vdots & \ddots & \vdots \\ y_{n1} & y_{n2} & \cdots & y_{np} \end{array}\right]}_{n\times p}$$
where n is the number of samples and p is the number of factors.

The principle component analysis of matrix (8) can be combined into *p* synthesis factors $y_{1},y_{2},\cdots y_{p},$ as follows:(9)$$\{\begin{array}{c} y_{1}=c_{11}y_{11}+c_{12}y_{12}+\cdots +c_{1p}x_{p}\\ y_{2}=c_{21}y_{21}+c_{22}y_{22}+\cdots +c_{2p}x_{p}\\ \cdots \cdots \\ y_{p}=c_{p1}y_{11}+c_{p2}y_{12}+\cdots +c_{pp}x_{p} \end{array}$$
where $c_{k1}^{2}+c_{k2}^{2}+\cdots +c_{kp}^{2}=1(k=1,2,\cdots p)$, and the comprehensive index factors $y_{1},y_{2},\cdots y_{p}$ is gradually reduced in variance.

According to Morrison, the main component should account for approximately 75% of the total variance. The relevant component is a parameter with an eigenvalue above 1 [34]. Based on this standard, four principle factors to make up the value of urban development quality are obtained. Furthermore, we use the method of efficacy coefficient to guarantee that all final scores are positive as follows: (10)$$y^{*}=\frac{y-\min(y)}{\max(y)-\min(y)}\times 0.4+0.6$$
where $y^{*}$ is the final value of urban development quality; y is the value of urban development quality calculated by PCA; min(y) is the minimum value of urban development quality calculated by PCA; and max(y) is the maximum value of urban development quality calculated by PCA.

To illustrate the spatial correlation of urban development quality in an intuitive way, the urban development quality of 285 prefecture-level cities under investigation in 2003 and 2016 are presented in Figure 2. Figure 2 shows two main observations. First, there are clear differences in urban development quality across three regions during 2003–2016, and urban development quality in the eastern, central and western regions is decreasing in turn except for 2013 and 2016. Second, urban development quality in the eastern, central and western regions has maintained an upward trend during 2003–2013, and there has been a slight decline in the three years after reaching the peak in 2013. The above results show that urban development quality in China has been fully upgraded in the past fourteen years, but the downward trend in the last three years should draw our constant attention.

### 3.2. Core Explanatory Variables

Limited to availability, continuity, and comparability of data, sulfur dioxide, and soot and dust are selected as the two main explanatory variables, and based on the difference of action stages, air pollution control is classified into two types: energy saving air pollution control and emission reduction air pollution control. Among them, energy saving air pollution control is the ratio of air pollutants discharged to local GDP, which indicates the cost of economic development; emission reduction air pollution control is the ratio of the removed amount discharged to the produced amount of air pollutants, which indicates the purification degree of air pollutants. To illustrate the spatial correlation of air pollution control in an intuitive way, the four indexes of air pollution control of 285 prefecture-level cities under investigation from 2003 to 2016 are presented in Figure 3, Figure 4, Figure 5 and Figure 6. Three main observations can be drawn from Figure 3, Figure 4, Figure 5 and Figure 6. First, the emission intensity of sulfur dioxide has declined rapidly while the emission intensity of smoke and dust has shown wave-like variation. Second, the removal rate of the two air pollutants has been increased rapidly, while the removal rate of smoke and dust is always higher than that of sulfur dioxide. These two pieces of evidence indicate that the control of smoke and dust is better than that of sulfur dioxide. Third, we have investigated that air pollution control in the eastern region is stronger than that in the central region and the western region during 2003–2016 on the whole, implying that local governments in the eastern region generally have deeper consciousness of environment protection and more sufficient supervision of the implement of air pollution control than those in the other two regions.

### 3.3. Control Variables

Specific to China, there are also some institutional and economic development factors contributing to the urban development quality in different regions in China. As a result, five control variables are included in the econometric estimation: (1) land finance (LF), i.e., the shares of land leasing revenue in GDP; (2) finance development (FD), i.e., the shares of both deposits and loans in GDP; (3) human capital (HC), i.e., the number of college students per 10,000 people; (4) foreign direct investment (FDI), i.e., the shares of foreign direct investment in GDP, and the annual exchange rate of RMB against the US dollars is used to convert FDI in US dollars to RMB; (5) industrial upgrading (IU), i.e., the shares of the value of the tertiary industries in the value of the secondary industries.

### 3.4. Descriptive Statistics and Correlation Coefficients for Regression Variables

To eliminate the impact of price fluctuations, with the year 2003 as the base period, the economic variables are processed at a constant price. To eliminate the influence of heteroscedasticity, this study has done logarithmic processing to all variables. Table 1 reports the data sources of relevant variables used in this paper. Appendix B also reports the data sources of relevant variables used in this paper. The correlation coefficients presented in Table 2 suggest significant and negative correlation between ln*ES* and ln*UDQ*, indicating the function of energy-saving air pollution control in blocking urban development quality in China; however, there is also a significant and positive correlation between ln*ER* and ln*UDQ*, indicating the role of emission-reduction air pollution control in promoting urban development quality in China.

## 4. Analysis and Discussion

### 4.1. Statistical Tests of Unit Root and Granger Causality

In the test of the unit root, we employed the methods of Levin et al. [36], Im et al. [37], and Maddala and Wu [38]. With an intercept and linear trend, each of these tests was carried out to include an intercept. As shown in Table 3 below, the unit root tests indicate that all the data series except ln*UDQ* and ln*HC* are static at a level, however, all the data series become static after the first difference is obtained.

Taking all factors into account, the existence of a unit root at the level and the absence of any at first difference is supported by the results of the unit root test. Additionally, results of both panel Granger causality and bootstrap Granger causality implied that Granger causal relationships run from ln*ES* and ln*ER* to ln*UDQ*, rather than bi-directionally. Thus, it is reasonable to further investigate the impact of air pollution control on urban development quality in China.

### 4.2. Estimation Results for the Whole Sample

In the use of spatial Durbin model, spatial dependence is investigated first. The results show that: the global Moran’s I index $w_{ij}^{1}$ is 0.202 and $w_{ij}^{2}$ is 0.194, both inconsistent with the original hypothesis at 1% significance level, indicating that the spatial econometric model should be selected for statistical verification using the maximum likelihood method. The LR test and the Wald test of spatial Durbin model (SDM) show that the original hypothesis is rejected at the 1% level of significance, i.e., spatial Durbin model (SDM) cannot degenerate into the spatial lag model (SLM) or spatial error model (SEM). The Hausman test result shows with a 1% significance level test, the fixed effect model of spatial Durbin model (SDM) should be selected. Further comprehensive analysis of the R_squared, the natural log-likelihood function value Log L, and the joint significance of LR test (space fixed and time fixed) reveal that spatial Durbin model (SDM) is more reasonable under the fixed effect of space-time. Hence, we choose the results of this model for analysis, and the results of the various model tests are shown in Table 4.

As can be seen in Table 4, the coefficients of four air pollution control indexes are consistent, indicating that the specification of the spatial weights matrices has no effect on the estimation results. Estimation results of the spatial Durbin model (SDM) in Table 4 show that ln*ES*_1_ has a significantly negative correlation with ln*UDQ*, while ln*ES*_2_ has a significantly positive correlation with ln*UDQ*. As discussed previously, the air pollution control during the sample period is mainly focused on smoke and dust emissions, while emissions of sulfur dioxide during the sample period are basically increasing, which is the evidence for the relationship between energy-saving air pollution control and urban development quality. However, the relationship between emission-reduction air pollution control and urban development quality is not significantly associated with ln*UDQ*, indicating that the effect of air pollution control on urban development quality is not satisfactory. Furthermore, the coefficients of ln*LF*, ln*HC* and ln*IU* also have a significantly positive association with ln*UDQ*, indicating the enhancement of land finance, human capital and industrial upgrading on urban development. Moreover, the coefficients of ln*FD* and ln*FDI* are also not significantly associated with ln*UDQ*, indicating that those two factors are weak in promoting urban development quality. As for the coefficients of the spatial item, only *w**ln*ES*_2_ has significantly positive correlation with ln*UDQ* and *w**ln*HC* is has a significantly negative correlation with it, while others are not significantly associated with ln*UDQ*, showing the importance of air pollution control in sulfur dioxide. In addition, the spatial coefficients (ρ) are also highly significant, which is a strong evidence of spatial dependence of urban development quality.

By introducing spatial effects into the traditional data model, the impact of air pollution control on urban development quality is no longer reflected only in the explanatory variables’ coefficient; instead, the spatial effect allows the impact to be disaggregated into direct and indirect effects.

As is shown in Table 5, the coefficients of direct effect and indirect effect are nearly consistent with the corresponding value in Table 4, indicating that the spatial feedback effects among different cities are negligible. The total effect of ln*ES*_1_ is significantly negatively correlated with ln*UDQ*, while ln*ES*_2_ is significantly positively correlated with ln*UDQ* and the other variables are not significant statistically, implying the importance of energy-saving air pollution control on urban development quality.

### 4.3. Estimation Results for the Sub-Regional Sample

China is a big country with vast territory. Therefore, the impact of air pollution control on urban development quality in different regions varies greatly. To take full account of the differences in urban development quality across regions, the regression is re-estimated using the sub-samples of three geographical regions (i.e., eastern, central and western) proposed by the National Bureau of Statistics (NBS). The prefecture-level cities in each region are listed in Appendix A. The estimation results for regression in these three regions are reported in Table 6.

Generally speaking, the results of three different regions are inconsistent with the results of the whole sample, which means the spatial heterogeneity of different regions is significant. Similar to the estimation results using the whole sample, the intensity of sulfur dioxide fails to play the expected role in promoting the increase of urban development quality, as the coefficients turn out to be negative (although not significant in the central region and western region). Furthermore, the intensity of smoke and dust are the core factor to improve the urban development quality, as the coefficients turn out to be positive (although not significant in the central region and western region). Moreover, the coefficients of ln*ER*_1_ and ln*ER*_2_ are significantly negatively and positively correlated with ln*UDQ* respectively in the eastern region, indicating that the difference of air pollution control on urban development quality is more prominent, in the eastern region which is the core of economic and industrial activities. Moreover, the coefficients of control variables in eastern region are similar to the estimation results of the whole sample except ln*FDI*, which means that FDI has played an important role in promoting the urban development quality in the eastern region. It is noteworthy that most of the coefficients are not significant in the central and western regions. As for the central region, only the coefficient of ln*HC* is significantly positively associated with ln*UDQ*, implying the importance of human capital in promoting the quality of urban development. As for the western region, only the coefficient of ln*LF* is significantly positively correlated with ln*UDQ*, implying the importance of land finance in promoting the quality of urban development. In addition, the spatial coefficients (ρ) are also highly significant in the eastern and western regions, but not significant in the central region, indicating the differentiation of spatial dependence in different regions.

Table 7 reports the direct, indirect and total effects of the eastern region. As is shown in Table 7, the coefficients of direct effect and indirect effect are nearly consistent with the corresponding coefficient in Table 6, indicating that the spatial feedback effects among eastern cities are also negligible. The total effect of ln*ES*_1_ has a significantly negative correlation with ln*UDQ*, while ln*ES*_2_ and ln*HC* have a significantly positive correlation with ln*UDQ*, while other variables are not statistically significant, implying the importance of energy-saving air pollution control and human capital on urban development quality in the eastern region.

Table 8 reports the direct, indirect and total effects of the central region. As is shown in Table 8, the coefficients of direct effect and indirect effect are nearly consistent with the corresponding coefficients in Table 6, indicating that the spatial feedback effects among central cities are also negligible. The total effect of ln*ER*_2_ is significantly and positively associated with ln*UDQ*, while other variables are not statistically significant, implying the importance of the removal rate of smoke and dust on urban development quality in the central region. It is noteworthy that most of the coefficients are not significant in the central region. One possible reason is that the intensity of air pollution control in the central region is relatively low compared to that in the eastern region. Besides, the local governments in the central region generally lack awareness of environment protection and do not have sufficient supervision on the implement of air pollution control.

Table 9 reports the direct, indirect and total effects of the western region. As is shown in Table 9, the coefficients of direct effect and indirect effect are nearly consistent with the corresponding coefficients in Table 6, indicating that the spatial feedback effects among western cities are also negligible. The total effect of ln*ER*_1_ is significantly and positively associated with ln*UDQ*, while other variables are not statistically significant, implying the importance of the intensity of sulfur dioxide on urban development quality in the western region. Similar to the central region, the intensity of air pollution control in the western region is relatively low compared to that in the eastern region. Besides, the local governments in the western region also lack consciousness of environment protection and do not have sufficient supervision on the implement of air pollution control. 

## 5. Conclusions and Policy Implications

With the panel data of 285 prefecture-level cities in China from 2003–2016, an index of urban development quality is constructed and calculated in this paper, based on the combination of principal component analysis and efficacy coefficient method. Through the application of spatial Durbin model, and under the framework of unified analysis, the role and mechanism of air pollution control on the impact of urban development quality are investigated. Three main conclusions can be drawn from the above analysis. First, during the investigation, the intensity of air pollution control and urban development quality have been enhanced, however, different air pollution control, especially the decline of sulfur dioxide, which erodes urban development quality at the national level, do not all play a positive role in improving the urban development quality as expected. Second, the impact of different types of air pollution control on urban development quality varies from region to region. In the eastern region, the direct effect of ln*ES*_1_ and ln*ER*_1_ is significantly and negatively correlated with urban development quality while the direct effect of ln*ES*_2_ and ln*ER*_2_ is significantly and positively associated with urban development quality, indicating that the control of smoke and dust has improved urban development quality, while the control of sulfur dioxide is at the sacrifice of the deterioration of urban development quality. In the central and the western regions, the direct effect of air pollution control on urban development quality does not pass the test of significance, indicating that the local government generally lacks consciousness of environment protection and does not have sufficient supervision of the implement of air pollution control compared with the local government in the eastern region. Third, the spatial coefficients (ρ) are also highly significant in China, the eastern region and the western region, which is a strong evidence of spatial dependence. However, it does not pass the test of significance in the central region, implying the spatial dependence of the central region is weak and poor compared with that of the eastern and western regions.

Three important policy implications can be drawn from the above conclusions. First, to comprehensively and thoroughly realize the goal of environmental protection and the improvement of urban development quality, it is necessary to promote the structural upgrading of the energy system. On the whole, clean energy should gradually replace fossil energy, for the reduction of smoke and dust will improve the urban development quality. However, if the alternative energy is insufficient, unreasonably pursuing the reduction of sulfur dioxide will hinder the improvement of urban development quality. Therefore, promoting the structural upgrading of the energy system is the most fundamental solution. Second, the local government should transform the excessive pursuit of short-term economic growth, and strive to improve the urban development quality comprehensively. Due to the lack of the consciousness of environment protection and the sufficient supervision of the implement of air pollution control, the impact of air pollution control on urban development quality in the central and western regions does not pass the test of significance, therefore, it is necessary and urgent to transform the reliance on extensive development models, and raise the awareness of environmental protection and sustainable development, and ultimately promote the improvement of urban development quality in the long run. Third, the links of environmental protection should be strengthened between different regions. Therefore, strengthening the legal basis of emission trading, compensation mechanisms and enforcement within the region and more importantly between regions can guarantee consistency and fairness of air pollution control policies, and ensure that the interests of urban development quality of inter-regional cities can be reasonably balanced.

Although this study provides valuable insights, it has three limitations, which should serve to stimulate further research. First, due to data restrictions, the period covered in this study is only fourteen years. To confirm our findings, the time span can be increased to cover a longer period, and more information and data can be used for comprehensive and thorough analysis. Second, in our study, air pollution control is divided into two types based on the difference before and after the treatment, and each type of air pollution control is measured by two typical indicators in empirical research. In further research, an expansion of the indicator system may be considered to obtain more guiding conclusions. Third, the spatial Durbin model is adopted to do the empirical analysis in this paper, but time effect is ignored, so the results may have some deviations compared to the actual situation. To expand the research, the dynamic spatial Durbin model should be adopted to empirically study the impact of air pollution control on urban development quality in China and other developing countries undergoing similar urbanization and modernization processes.

## Figures and Tables

**Figure 1 ijerph-15-02822-f001:**
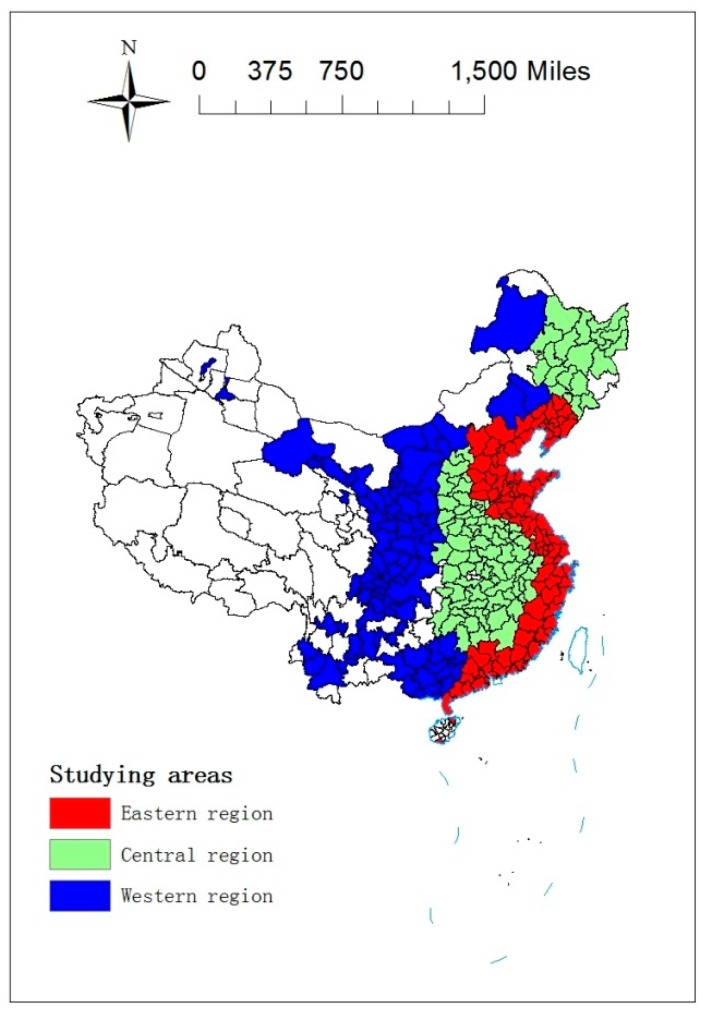
Studying areas.

**Figure 2 ijerph-15-02822-f002:**
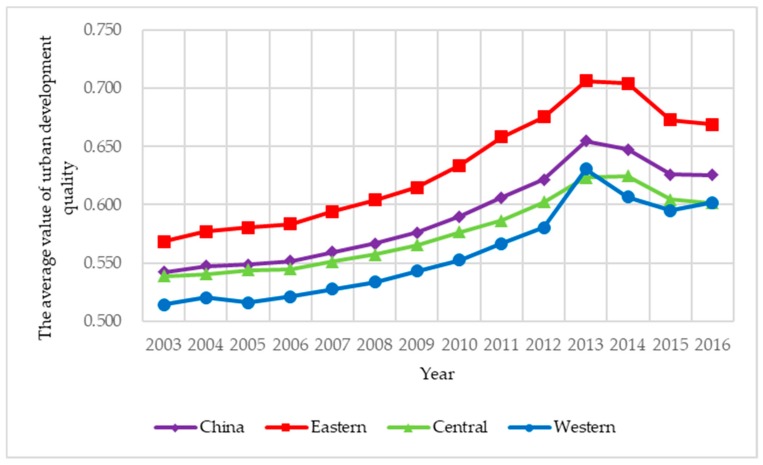
The average value of urban development quality in China during 2003–2016.

**Figure 3 ijerph-15-02822-f003:**
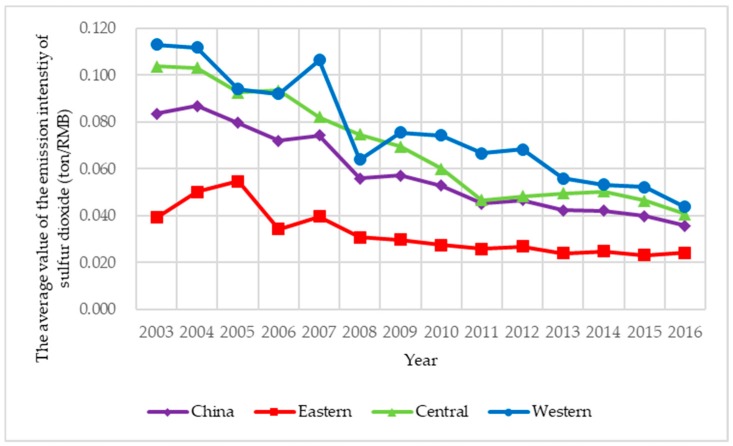
The average value of the emission intensity of sulfur dioxide in China during 2003–2016.

**Figure 4 ijerph-15-02822-f004:**
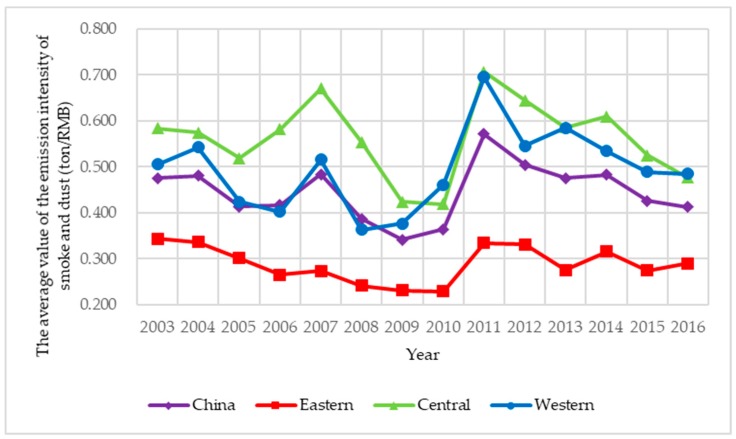
The average value of the emission intensity of smoke and dust in China during 2003–2016.

**Figure 5 ijerph-15-02822-f005:**
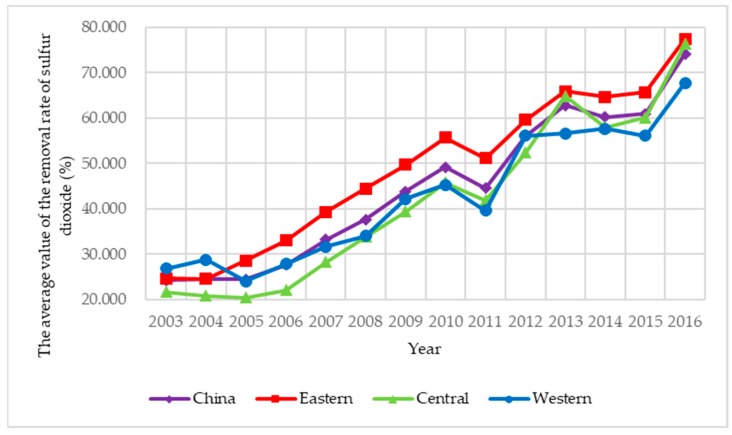
The average value of the removal rate of sulfur dioxide in China during 2003–2016.

**Figure 6 ijerph-15-02822-f006:**
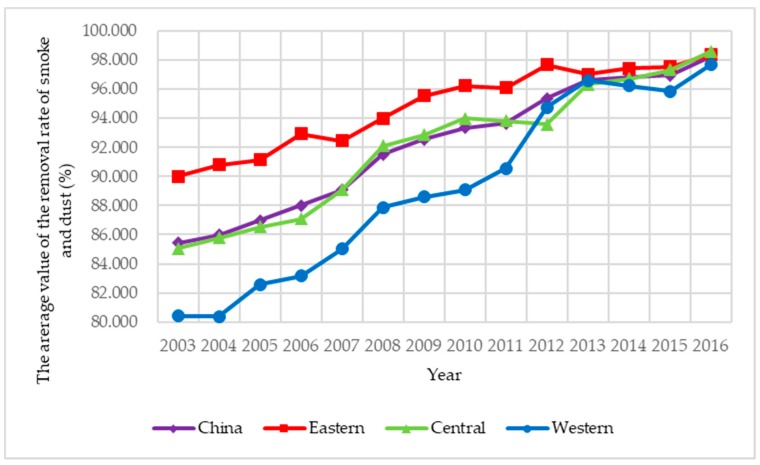
The average value of the removal rate of smoke and dust in China during 2003–2016.

**Table 1 ijerph-15-02822-t001:** Descriptive statistics.

Variables	Definition	Obs.	Unit	Std. Dev.	Mean	Min	First Quartile	Median Quartile	Third Quartile	Max	Kurtosis	Skewness
ln*UDQ*	Urban development quality	3990	-	0.185	−0.536	−1.047	−0.647	−0.570	−0.464	2.257	41.015	4.005
ln*ES*_1_	The emission intensity of sulfur dioxide per GDP	3990	ton/RMB	1.356	−3.734	−14.514	−4.481	−3.707	−2.878	1.334	3.734	−0.642
ln*ES*_2_	The emission intensity of smoke and dust per GDP	3990	ton/RMB	1.690	−1.755	−12.329	−2.673	−1.468	−0.619	1.816	2.869	−1.193
ln*ER*_1_	The removal rate of sulfur dioxide	3990	%	1.159	−1.194	−8.517	−1.623	−0.777	−0.414	0.000	5.158	−1.962
ln*ER*_2_	The removal rate of smoke and dust	3990	%	0.309	−0.109	−6.908	−0.067	−0.026	−0.012	0.000	101.414	−7.942
ln*LF*	The shares of land leasing revenue in GDP	3990	%	1.536	−4.928	−13.119	−5.743	−4.408	−4.122	−1.419	1.924	−1.099
ln*FD*	The shares of both deposits and loans in GDP	3990	%	0.763	1.308	−4.838	0.830	1.559	1.732	4.605	10.033	0.229
ln*HC*	The number of college students per 10,000 people	3990	10^4^ persons	1.174	0.940	−9.210	0.376	1.179	1.653	3.177	6.36	−1.696
ln*FDI*	The shares of foreign direct investment in GDP	3990	%	2.954	−2.922	−12.512	−4.682	−3.693	−2.449	4.605	0.311	0.879
ln*IU*	The shares of the value of the tertiary industries in the value of the secondary industries	3990	%	0.517	−0.172	−2.361	−0.484	−0.177	0.137	1.621	1.377	−0.073

**Table 2 ijerph-15-02822-t002:** Correlation coefficients for regression variables.

	ln*UDQ*	ln*ES*_1_	ln*ES*_2_	ln*ER*_1_	ln*ER*_2_	ln*LF*	ln*FD*	ln*HC*	ln*FDI*	ln*IU*
ln*UDQ*	1.000									
ln*ES*_1_	−0.250 ***	1.000								
ln*ES*_2_	−0.101 ***	0.684 ***	1.000							
ln*ER*_1_	0.245 ***	0.063 ***	0.078 ***	1.000						
ln*ER*_2_	0.166 ***	0.073 ***	0.356 ***	0.202 ***	1.000					
ln*LF*	0.151 ***	−0.010	0.004	0.080 ***	−0.035 **	1.000				
ln*FD*	0.170 ***	0.013	0.070 ***	0.169 ***	0.040 *	0.150 ***	1.000			
ln*HC*	0.321 ***	−0.039 **	0.011	0.114 ***	0.071 ***	0.094 ***	0.024	1.000		
ln*FDI*	−0.027 *	0.044 ***	−0.014	−0.017	−0.048 ***	0.010	0.068 ***	−0.121 ***	1.000	
ln*IU*	0.074 ***	−0.098 ***	−0.11 ***	0.043 ***	−0.091 ***	0.148 ***	0.173 ***	0.094 ***	0.120 ***	1.000

Note: ***, **, and * denote statistical significance at the 1%, 5%, and 10% significance levels, respectively.

**Table 3 ijerph-15-02822-t003:** Panel unit root test results.

Variables	Level			First Difference		
	Levin, Liu and Chu	Im, Pesaran and Shin	ADF	Levin, Liu and Chu	Im, Pesaran and Shin	ADF
ln*UDQ*	−14.570 ***	−1.097	635.810 **	−13.066 ***	−11.994 ***	1134.650 ***
ln*ES*_1_	−24.228 ***	−5.974 ***	780.801 ***	−32.4714 ***	−17.686 ***	1323.270 ***
ln*ES*_2_	−58.479 ***	−12.867 ***	923.438 ***	−57.117 ***	−12.728 ***	1594.850 ***
ln*ER*_1_	−13.115 ***	−3.601 ***	708.664 ***	−25.195 ***	−15.038 ***	1189.290 ***
ln*ER*_2_	−16.644 ***	−4.461 ***	727.268 ***	−26.565 ***	−15.395 ***	1205.780 ***
ln*LF*	−12.378 ***	−4.888 ***	754.713 ***	−24.878 ***	−17.481 ***	1275.580 ***
ln*FD*	−9.233 ***	−5.049 ***	699.444 ***	−12.991 ***	−15.432 ***	1203.480 ***
ln*HC*	0.084	5.779	533.904	−5.528 ***	−10.172 ***	1116.980 ***
ln*FDI*	−40.262 ***	−8.937 ***	844.094 ***	−99.703 ***	−59.857 ***	3159.320 ***
ln*IU*	−11.750 ***	−0.971	672.209 ***	−25.091 ***	−17.406 ***	1310.790 ***

Notes: *** and ** denote statistical significance at the 1% and 5% significance levels, respectively.

**Table 4 ijerph-15-02822-t004:** The results for the whole sample.

Variables	$w_{ij}^{1}$				$w_{ij}^{2}$			
Constant	−0.470 ***				−0.464 ***			
	(−20.333)				(−19.795)			
ln*ES*_1_	−0.026 ***	−0.019 ***	−0.023 ***	−0.009 ***	−0.026 ***	−0.018 ***	−0.024 ***	−0.009 ***
	(−9.378)	(−5.430)	(−8.069)	(−2.635)	(−9.619)	(−5.270)	(−8.428)	(−2.622)
ln*ES*_2_	−0.005 **	0.010 ***	−0.007 ***	0.006 ***	−0.005**	0.010 ***	−0.007 ***	0.006 ***
	(−2.177)	(4.061)	(−3.145)	(2.694)	(−2.325)	(3.986)	(−3.177)	(2.721)
ln*ER*_1_	0.017 ***	0.007 ***	0.014 ***	−0.002	0.017 ***	0.007 ***	0.014 ***	−0.002
	(7.224)	(2.849)	(5.639)	(−0.869)	(7.133)	(2.709)	(5.777)	(−0.801)
ln*ER*_2_	0.066 ***	0.006	0.069 ***	0.003	0.066 ***	0.006	0.070 ***	0.003
	(7.372)	(0.743)	(7.771)	(0.323)	(7.447)	(0.680)	(7.897)	(0.330)
ln*LF*	0.009 ***	0.007 ***	0.008 ***	0.005 ***	0.008 ***	0.007 ***	0.008 ***	0.005 ***
	(5.383)	(4.241)	(5.019)	(2.987)	(5.251)	(4.215)	(4.902)	(2.981)
ln*FD*	0.030 ***	0.006	0.027 ***	0.004	0.031 ***	0.006	0.029 ***	0.004
	(6.183)	(1.286)	(5.742)	(0.799)	(6.539)	(1.314)	(6.080)	(0.781)
ln*HC*	0.047 ***	0.015 ***	0.047 ***	0.012 ***	0.048 ***	0.016 ***	0.048 ***	0.012 ***
	(19.616)	(4.495)	(19.680)	(3.656)	(20.120)	(4.778)	(20.103)	(3.745)
ln*FDI*	0.001	0.001	0.001	0.001	0.001	0.001	0.001	0.001
	(1.001)	(0.567)	(1.238)	(1.041)	(0.650)	(0.618)	(0.898)	(1.025)
ln*IU*	−0.008	0.022 ***	−0.008	0.019 ***	−0.009	0.021 ***	−0.008 *	0.019 ***
	(−1.604)	(3.062)	(−1.542)	(2.659)	(−1.912)	(2.994)	(−1.685)	(2.707)
*w**n*ES*_1_	−0.013 ***	−0.034 ***	−0.004	−0.003	−0.011 ***	−0.035 ***	−0.004	−0.005
	(−3.277)	(−7.051)	(−0.966)	(−0.614)	(−2.936)	(−7.354)	(−0.943)	(−0.922)
*w**ln*ES*_2_	0.010 ***	0.018 ***	0.004	0.007 *	0.011 ***	0.018 ***	0.006	0.008**
	(3.224)	(4.634)	(1.166)	(1.912)	(3.510)	(4.676)	(1.624)	(2.121)
*w**ln*ER*_1_	0.017 ***	0.019 ***	0.010 ***	−0.001	0.018 ***	0.019 ***	0.012 ***	−0.002
	(5.311)	(6.042)	(2.701)	(−0.375)	(5.448)	(5.741)	(3.053)	(−0.635)
*w**ln*ER*_2_	0.001	−0.001	0.012	−0.005	0.001	−0.004	0.013	−0.006
	(0.069)	(−0.083)	(0.863)	(−0.398)	(0.071)	(−0.285)	(0.875)	(−0.45)
*w**ln*LF*	0.003	0.002	0.001	−0.003	0.002	0.002	0.001	−0.002
	(1.051)	(0.828)	(0.544)	(−1.211)	(0.907)	(0.96)	(0.535)	(−0.912)
*w**ln*FD*	−0.016 ***	0.008	−0.023 ***	−0.004	−0.464 ***	0.000	0.000	0.000
	(−2.897)	(1.500)	(−3.906)	(−0.667)	(−19.795)	(0.000)	(0.000)	(0.000)
*w**ln*HC*	−0.018 ***	−0.008 **	−0.015 ***	−0.013 ***	−0.026 ***	−0.018 ***	−0.024 ***	−0.009 ***
	(−5.743)	(−2.017)	(−4.693)	(−3.329)	(−9.619)	(−5.270)	(−8.428)	(−2.622)
*w**ln*FDI*	0.000	−0.002	0.001	−0.001	−0.005 **	0.010 ***	−0.007 ***	0.006 ***
	(0.030)	(−1.583)	(0.555)	(−0.804)	(−2.325)	(3.986)	(−3.177)	(2.721)
*w**ln*IU*	0.022 ***	0.001	0.023 ***	−0.003	0.017 ***	0.007 ***	0.014 ***	−0.002
	(3.086)	(0.109)	(3.190)	(−0.303)	(7.133)	(2.709)	(5.777)	(−0.801)
*ρ*	0.257 ***	0.208 ***	0.234 ***	0.126 ***	0.066 ***	0.006	0.070 ***	0.003
	(15.552)	(12.261)	(13.927)	(7.105)	(7.447)	(0.680)	(7.897)	(0.330)
Space-fixed	No	Yes	No	Yes	No	Yes	No	Yes
Time-fixed	No	No	Yes	Yes	No	No	Yes	Yes
R-squared	0.330	0.567	0.345	0.594	0.330	0.567	0.345	0.594
Log-likelihood	1825.446	2708.772	1876.033	2860.128	1826.725	2709.785	1877.126	2860.577
Moran’s I	0.202 ***				0.194 ***			
LR_joint_space fixed	1112.135 ***				1126.444 ***			
LR_joint_time fixed	277.543 ***				277.030 ***			
Wald_spatial_lag	17.416 **				18.580 **			
LR_spatial_lag	18.673 **				20.004 **			
Wald_spatial_error	16.498 **				18.035 **			
LR_spatial_error	17.810 **				19.471 **			
Hauman test	1066.035 ***				1208.316 ***			
obs	3990	3990	3990	3990	3990	3990	3990	3990

Notes: The t-statistics are given in the parentheses; ***, **, and * denote statistical significance at the 1%, 5%, and 10% significance levels, respectively.

**Table 5 ijerph-15-02822-t005:** The direct, indirect and total effects of the whole sample.

Variables	$w_{ij}^{1}$			$w_{ij}^{2}$		
	Direct Effect	Indirect Effect	Total Effect	Direct Effect	Indirect Effect	Total Effect
ln*ES*_1_	−0.010 ***	−0.005	−0.014 **	−0.009 ***	−0.007	−0.016 **
	(−2.768)	(−0.755)	(−2.003)	(−2.815)	(−1.108)	(−2.324)
ln*ES*_2_	0.007 ***	0.009 **	0.016 ***	0.007 ***	0.010 **	0.017 ***
	(2.800)	(2.037)	(3.042)	(2.824)	(2.238)	(3.298)
ln*ER*_1_	−0.002	−0.002	−0.004	−0.002	−0.003	−0.005
	(−0.934)	(−0.422)	(−0.815)	(−0.812)	(−0.744)	(−1.020)
ln*ER*_2_	0.002	−0.006	−0.003	0.002	−0.007	−0.005
	(0.271)	(−0.388)	(−0.185)	(0.282)	(−0.480)	(−0.259)
ln*LF*	0.005 ***	−0.003	0.002	0.005 ***	−0.002	0.003
	(2.966)	(−0.915)	(0.647)	(2.876)	(−0.655)	(0.912)
ln*FD*	0.004	−0.004	0.000	0.004	−0.003	0.000
	(0.816)	(−0.640)	(−0.022)	(0.802)	(−0.548)	(0.088)
ln*HC*	0.011 ***	−0.013 ***	−0.001	0.011 ***	−0.013 ***	−0.002
	(3.420)	(−3.209)	(−0.382)	(3.596)	(−3.282)	(−0.439)
ln*FDI*	0.001	−0.001	0.000	0.001	−0.001	0.000
	(0.980)	(−0.706)	(−0.064)	(1.028)	(−0.658)	(−0.012)
ln*IU*	0.019 ***	0.000	0.019	0.019 ***	−0.001	0.019
	(2.652)	(0.003)	(1.363)	(2.700)	(−0.053)	(1.332)

Notes: The t-statistics are given in the parentheses; *** and ** denote statistical significance at the 1% and 5% significance levels, respectively.

**Table 6 ijerph-15-02822-t006:** The results of the sub-regional sample.

Variables	Eastern Region		Central Region		Western Region	
	$w_{ij}^{1}$	$w_{ij}^{2}$	$w_{ij}^{1}$	$w_{ij}^{2}$	$w_{ij}^{1}$	$w_{ij}^{2}$
ln*ES*_1_	−0.016 ***	−0.016 ***	−0.006	−0.005	−0.003	−0.003
	(−3.088)	(−3.096)	(−0.575)	(−0.492)	(−0.731)	(−0.744)
ln*ES*_2_	0.007 *	0.007 *	0.008	0.009	0.001	0.001
	(1.755)	(1.799)	(1.413)	(1.478)	(0.469)	(0.433)
ln*ER*_1_	−0.010 *	−0.010 *	−0.002	−0.002	0.002	0.002
	(−1.779)	(−1.756)	(−0.422)	(−0.381)	(0.931)	(0.912)
ln*ER*_2_	0.042 **	0.042 **	0.018	0.017	−0.007	−0.007
	(2.071)	(2.065)	(0.822)	(0.769)	(−0.962)	(−0.946)
ln*LF*	0.006 **	0.006 **	0.003	0.003	0.004**	0.004 **
	(2.104)	(2.081)	(1.040)	(1.037)	(2.296)	(2.264)
ln*FD*	−0.006	−0.006	0.009	0.007	−0.004	−0.004
	(−0.712)	(−0.691)	(0.798)	(0.654)	(−0.687)	(−0.583)
ln*HC*	0.041 ***	0.041 ***	0.021 **	0.021 **	−0.001	−0.001
	(5.469)	(5.465)	(2.560)	(2.506)	(−0.329)	(−0.322)
ln*FDI*	0.007 **	0.007 **	−0.002	−0.001	0.001	0.001
	(2.192)	(2.094)	(−0.679)	(−0.594)	(0.973)	(0.968)
ln*IU*	0.036 **	0.038 **	0.013	0.013	0.007	0.007
	(2.257)	(2.338)	(0.862)	(0.912)	(0.895)	(0.955)
*w**ln*ES*_1_	−0.009	−0.011	0.021	0.012	0.005	0.006
	(−1.077)	(−1.311)	(1.401)	(0.798)	(0.753)	(0.947)
*w**ln*ES*_2_	0.014 **	0.013 **	0.002	0.005	0.004	0.004
	(2.241)	(2.181)	(0.181)	(0.600)	(0.972)	(0.94)
*w**ln*ER*_1_	−0.004	−0.002	−0.008	−0.009	0.007 *	0.007 *
	(−0.460)	(−0.28)	(−1.186)	(−1.325)	(1.947)	(1.702)
*w**ln*ER*_2_	−0.042	−0.042	0.059 **	0.057 *	−0.014	−0.009
	(−1.085)	(−0.998)	(2.118)	(1.798)	(−1.09)	(−0.753)
*w**ln*LF*	−0.002	−0.001	−0.005	−0.004	0.001	0.001
	(−0.437)	(−0.122)	(−0.973)	(−0.746)	(0.416)	(0.310)
*w**ln*FD*	−0.008	−0.006	−0.008	−0.002	0.003	0.002
	(−0.608)	(−0.489)	(−0.444)	(−0.105)	(0.394)	(0.271)
*w**ln*HC*	−0.016	−0.017	−0.036 ***	−0.034 ***	0.004	0.004
	(−1.549)	(−1.550)	(−3.048)	(−2.997)	(0.946)	(0.874)
*w**ln*FDI*	−0.006 *	−0.006	−0.003	−0.004	0.000	0.001
	(−1.645)	(−1.513)	(−1.056)	(−1.056)	(0.223)	(0.391)
*w**ln*IU*	−0.036	−0.036	−0.032	−0.032	−0.006	−0.001
	(−1.423)	(−1.459)	(−1.437)	(−1.471)	(−0.588)	(−0.125)
*ρ*	0.200 ***	0.209 ***	0.026	0.032	0.058 *	0.054 *
	(7.215)	(7.565)	(0.875)	(1.058)	(1.869)	(1.728)
Space-fixed	Yes	Yes	Yes	Yes	Yes	Yes
Time-fixed	Yes	Yes	Yes	Yes	Yes	Yes
R-squared	0.597	0.597	0.489	0.488	0.694	0.694
Log-likelihood	974.957	974.424	820.675	820.077	1297.524	1296.937
Moran’s I	0.247 ***	0.240 ***	0.076 ***	0.066 ***	0.127 ***	0.124 ***
LR_joint_space fixed	282.001 ***	284.884 ***	354.704 ***	355.597 ***	464.838 ***	465.884 ***
LR_joint_time fixed	111.213 ***	111.311 ***	93.872 ***	94.043 ***	162.776 ***	157.463 ***
Wald_spatial_lag	14.646	13.997	19.022 **	17.887 **	9.606	8.810
LR_spatial_lag	15.740 *	15.059 *	20.461 **	19.344 **	10.514	9.967
Wald_spatial_error	12.960	12.702	18.940 **	17.812 **	9.883	9.060
LR_spatial_error	14.160	13.829	20.394 **	19.311 **	10.771	9.880
Hauman test	208.849 ***	153.087 ***	26.768 *	61.305 ***	336.721 ***	378.552 ***
obs	1414	1414	1400	1400	1176	1176

Notes: The t-statistics are given in the parentheses; ***, **, and * denote statistical significance at the 1%, 5%, and 10% significance levels, respectively.

**Table 7 ijerph-15-02822-t007:** The direct, indirect and total effects of the eastern region.

Variables	$w_{ij}^{1}$			$w_{ij}^{2}$		
	Direct Effect	Indirect Effect	Total Effect	Direct Effect	Indirect Effect	Total Effect
ln*ES*_1_	−0.017 ***	−0.014	−0.031 ***	−0.017 ***	−0.016 *	−0.033 ***
	(−3.395)	(−1.447)	(−2.730)	(−3.318)	(−1.705)	(−2.993)
ln*ES*_2_	0.008 **	0.018 **	0.026 ***	0.009 **	0.018 **	0.026 ***
	(2.033)	(2.472)	(2.928)	(2.109)	(2.512)	(2.959)
ln*ER*_1_	−0.010 *	−0.007	−0.017	−0.010 *	−0.005	−0.015
	(−1.751)	(−0.703)	(−1.329)	(−1.754)	(−0.447)	(−1.123)
ln*ER*_2_	0.041 *	−0.040	0.001	0.040 *	−0.039	0.000
	(1.886)	(−0.850)	(0.009)	(1.831)	(−0.787)	(0.002)
ln*LF*	0.006 **	−0.001	0.005	0.006 **	0.001	0.007
	(2.053)	(−0.183)	(0.859)	(2.051)	(0.158)	(1.111)
ln*FD*	−0.007	−0.010	−0.018	−0.007	−0.009	−0.016
	(−0.825)	(−0.711)	(−1.069)	(−0.767)	(−0.637)	(−0.958)
ln*HC*	0.040 ***	−0.010	0.030 **	0.040 ***	−0.009	0.031 **
	(5.446)	(−0.818)	(2.257)	(5.444)	(−0.762)	(2.226)
ln*FDI*	0.007 **	−0.006	0.001	0.007 **	−0.006	0.001
	(2.314)	(−1.566)	(0.195)	(2.281)	(−1.293)	(0.254)
ln*IU*	0.034 **	−0.033	0.001	0.036 **	−0.034	0.002
	(2.155)	(−1.155)	(0.034)	(2.227)	(−1.172)	(0.062)

Notes: The t-statistics are given in the parentheses; ***, **, and * denote statistical significance at the 1%, 5%, and 10% significance levels, respectively.

**Table 8 ijerph-15-02822-t008:** The direct, indirect and total effects of the central region.

Variables	$w_{ij}^{1}$			$w_{ij}^{2}$		
	Direct Effect	Indirect Effect	Total Effect	Direct Effect	Indirect Effect	Total Effect
ln*ES*_1_	−0.006	0.022	0.016	−0.005	0.013	0.008
	(−0.582)	(1.463)	(0.936)	(−0.467)	(0.851)	(0.494)
ln*ES*_2_	0.008	0.002	0.010	0.009	0.005	0.014
	(1.481)	(0.221)	(0.974)	(1.534)	(0.582)	(1.334)
ln*ER*_1_	−0.002	−0.009	−0.011	−0.002	−0.010	−0.012
	(−0.436)	(−1.236)	(−1.274)	(−0.413)	(−1.301)	(−1.329)
ln*ER*_2_	0.018	0.060 **	0.079 **	0.016	0.059 *	0.075 *
	(0.873)	(2.149)	(2.217)	(0.764)	(1.778)	(1.900)
ln*LF*	0.003	−0.005	−0.002	0.003	−0.003	0.000
	(0.995)	(−1.000)	(−0.258)	(1.054)	(−0.683)	(0.016)
ln*FD*	0.009	−0.008	0.001	0.007	−0.001	0.006
	(0.813)	(−0.426)	(0.0600)	(0.633)	(−0.083)	(0.292)
ln*HC*	0.021 **	−0.036 ***	−0.015	0.020 **	−0.035 ***	−0.014
	(2.448)	(−3.029)	(−1.072)	(2.475)	(−2.931)	(−1.053)
ln*FDI*	−0.002	−0.004	−0.005	−0.002	−0.004	−0.005
	(−0.685)	(−1.033)	(−1.436)	(−0.613)	(−1.122)	(−1.460)
ln*IU*	0.012	−0.033	−0.021	0.013	−0.034	−0.021
	(0.812)	(−1.439)	(−0.840)	(0.884)	(−1.537)	(−0.851)

Notes: The t-statistics are given in the parentheses; ***, **, and * denote statistical significance at the 1%, 5%, and 10% significance levels, respectively.

**Table 9 ijerph-15-02822-t009:** The direct, indirect and total effects of the western region.

Variables	$w_{ij}^{1}$			$w_{ij}^{2}$		
	Direct Effect	Indirect Effect	Total Effect	Direct Effect	Indirect Effect	Total Effect
ln*ES*_1_	−0.003	0.005	0.001	−0.003	0.006	0.003
	(−0.718)	(0.701)	(0.168)	(−0.778)	(0.898)	(0.328)
ln*ES*_2_	0.002	0.005	0.006	0.001	0.005	0.006
	(0.554)	(0.965)	(1.089)	(0.482)	(0.976)	(1.049)
ln*ER*_1_	0.003	0.008 *	0.010 **	0.002	0.007 *	0.010 *
	(0.961)	(1.940)	(2.127)	(0.927)	(1.784)	(1.939)
ln*ER*_2_	−0.008	−0.014	−0.022	−0.007	−0.010	−0.017
	(−1.018)	(−1.084)	(−1.379)	(−0.993)	(−0.789)	(−1.162)
ln*LF*	0.004 **	0.002	0.006 *	0.004 **	0.001	0.005
	(2.422)	(0.530)	(1.723)	(2.371)	(0.364)	(1.531)
ln*FD*	−0.005	0.003	−0.002	−0.004	0.003	−0.002
	(−0.714)	(0.395)	(−0.214)	(−0.634)	(0.304)	(−0.226)
ln*HC*	−0.001	0.004	0.003	−0.001	0.004	0.003
	(−0.335)	(0.994)	(0.676)	(−0.268)	(0.891)	(0.627)
ln*FDI*	0.001	0.000	0.001	0.001	0.001	0.002
	(0.955)	(0.295)	(0.810)	(1.018)	(0.399)	(0.902)
ln*IU*	0.007	−0.006	0.001	0.008	−0.001	0.007
	(0.910)	(−0.569)	(0.049)	(0.977)	(−0.071)	(0.472)

Notes: The t-statistics are given in the parentheses; ** and * denote statistical significance at the 5% and 10% significance levels, respectively.

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
