# Peer review of "Effects of Air Pollution Control on Urban Development Quality in Chinese Cities Based on Spatial Durbin Model"

_ijerph, 2018, doi:10.3390/ijerph15122822_

Round 1
Reviewer 1 Report
Manuscript: Effects of Air Pollution Control on Urban Development Quality in Chinese Cities Based on Spatial Durbin Model
Ref: IJERPH 383015
Background
The manuscript presents an interesting approach for assessing urban development trends in China for a period of 14 years.
However, the manuscript has several shortcomings that require improvements and corrections to comply with the scientific requirements of IJERPH, a high impact and reputed international journal.
The author must improve the structure of this manuscript, the content and some technical issues that need to be addressed. Consequently, I would recommend major revision because of poor readability.
IMPORTANT ISSUES
I suggest a better and more narrowed presentation of the data presented in the tables and graphs making more correlations of the results and adequate discussions related to latest references. Try to extract and discuss more information from tables and graphs.
Suggestions to improve the study and to increase the complexity of the approach:
1. IMPORTANT! I suggest the authors to use a professional proofing service or have their revised manuscript checked by a native English speaker in the field of air pollution/statistics research. There are many grammar and formulation issues (ambiguous phrases, use of definite article; typos; punctuation mistakes e.g., use of comma, space, brackets etc.) that diminishes the quality of the paper and most important: the readability. Verbose is significant in this manuscript.
2. The authors should read, edit and check several times their manuscript before resubmitting to avoid typos and punctuation drawbacks.
3. Regarding the maps, for the comparison purpose, it should be better to use the same categories for both years 2013 and 2016. In this way, differences could be observed.
4. Try to include a graph that presents the trend of a relevant parameter related to UDQ for the whole time series (2013-2016) for each of the three regions. This will present clearly the general pattern.
5. Include statistics related to dispersion such as skewness and kurtosis as well as for central tendency (median and interquartilic range) – e.g. table 1
6. Explain why you choose 10% level as the latest significant threshold instead of 5%. What are the risks? (L236, L290 and so on).
7. Avoid unusual terms such as spurious (L239), pale (L275) with scientific ones.
8. Abstract should be revised according to the recommendation from IJERPH journal
“The abstract should be a total of about 200 words maximum. The abstract should be a single paragraph and should follow the style of structured abstracts, but without headings: 1) Background: Place the question addressed in a broad context and highlight the purpose of the study; 2) Methods: Describe briefly the main methods or treatments applied. Include any relevant preregistration numbers, and species and strains of any animals used. 3) Results: Summarize the article's main findings; and 4) Conclusion: Indicate the main conclusions or interpretations. The abstract should be an objective representation of the article: it must not contain results which are not presented and substantiated in the main text and should not exaggerate the main conclusions.”
SPECIFIC COMMENTS (this is a non-exhaustive list):
L12 the term smoke (powder) dust is not common in atmospheric and air pollution studies. More correct should be smoke and dust. Powder is related to particulate matter pollution. The term should be explained to avoid confusions.
L15 is significant in statistical tests…
L24 All the phrases in this manuscript should be re-checked and some of them must be shortened e.g., . However, … The same for L29. Reformulate L44-46
L59-60 this statement is probably available for Chinese literature referring to China and it should be better to mention this.
L64 is another statement that requires evidence. In my opinion should be deleted or reformulated
L82 an example of space misuse (space before This…) There are many mistakes and should be checked throughout the manuscript.
L105 Check the rectangle (maybe a character mistake).
L108 should be corrected
L198 replace government with management or control
L201 has not have
Table 1 – typo = variable
L241 include relevant reference for LLC approach that explains the methodology
L314 has played
L363 – typo = penal (maybe panel)
L365 efficacy coefficient method
Conclusions should be short sentences providing the key findings, limitations and future work.
The current form is an abstract of the manuscript repeating information presented in the previous sections. Most of the information presented should be more adequate to discussions. This section is poorly presented in the manuscript.
Author Response
Dear Reviewer,
We would like to thank you for your interest in our study and for the constructive comments. We have made the following revisions accordingly.
1. We have used a professional proofing service and have our revised manuscript checked by two native English speakers in the field of air pollution/statistics research to correct the grammar and format issues.
2. We have read, edited and checked our revision several times before resubmitting it to avoid spelling and punctuation mistakes.
3. Taking into account the obstacles of original graphics in the contrast analysis, we replaced it with a new map and marked the study areas in it.
4. We have used line charts instead of maps and compared the average value of the dependent variables with the main explanatory variables among different regions. In this way, you can clearly identify trends of different variables in different regions.
5. We have added statistics related to dispersion such as skewness and kurtosis as well as central tendency (median and interquartilic range), and we appreciate your advice.
6. This standard is used in many existing literatures, and we continue with their practices of our predecessors, which not only expands the range of stability between variables, but also complies with the implementation of 5% standards.
7. We have minimized the use of these unusual terms through multiple rounds of screening and replacement.
8. The abstract have been revised according to the recommendations from IJERPH journal, including backgrounds, methods, results and conclusions.
Above are the detailed corrections we made according to your comments point by point. Thanks again for your meticulous review and valuable suggestions to improve our manuscript. We hope that the revision has addressed all the issues in the old version. However, in the event that the modifications listed above DO NOT meet your expectations, we sincerely request you to give us another opportunity to revise the paper. We are very grateful for this opportunity given to us and would be more than happy to revise the paper until you are satisfied. We are looking forward to your positive response.
Best regards,
Yanchao Feng Xiaohong Wang* Wenchao Du Jun Liu
Reviewer 2 Report
This study is about the air pollution control impact in Chinese cities. The main conclusions are the impact of the air pollution control on urban development in the eastern-coastal region is significant statistically, and the impact is not significance over the central and western region of china. I have some detail comments listed below.
1, Please specify the ‘smoke (powder) dust’, it is confusing. Does this mean the general particular matter (PM)? Does this include all the aerosols in the atmosphere no matter they are from anthropogenic emission or from some natural sources?
2, This article studied sulfur dioxide, how about another important component – Nitrogen dioxide?
3, This study is about air pollution control, so mainly control anthropogenic emissions? Is there smoke from fire? What are the uncertainties for this study, regarding some possible natural emission sources such as dust, and forest fire smoke? How did you consider the man made biomass burning?
4, Line 17, SDM, please specify the meaning here.
5, For the figures, Figure 1 – Figure5, the scale of the color bars should be same for both the year of 2003 and 2016. Or it is very hard to visualize the difference between these two years. And also what is the unit for ‘emission of sulfur dioxide per GDP’, what is the unit for ‘smoke (powder) dust’, the removal rate of ‘sulfur dioxide’, ‘smoke …’, please either specify the unit on the figure or in the figure caption.
6, Line 240, need to introduce some necessary basic information of ‘Levin-Lin-Chiu’ approach. Did any previous studies use this approach and also do some verification of the results? How reliable of this approach?
7, Are there similar study with similar/different conclusions? It is good to address and compare against the conclusions from some literatures to make the results convincing.
Author Response
Dear Reviewer,
We would like to thank you for your interest in our study and for the constructive comments. We have made the following revisions accordingly.
1. In keeping with the existing literature, ‘smoke (powder) dust’ has been corrected to ‘smoke and dust’ , because in the book China City Statistical Yearbook, the index of smoke and dust refers to the amount of industrial smoke and dust, not including others.
2. We’d like to add nitrogen dioxide to enrich our research, but we can’t find the data at the prefecture level. As we have mentioned in our outlook, with the availability of data, the increase in relevant indicators will improve the reliability and practicality, and we will continue to track it.
3. As the answer to the first question, the air pollution variables used in this paper are all industrial indicators.
4. SDM is the abbreviation of spatial Durbin model, and we have corrected all similar abbreviations.
5. We have used line charts instead of maps and compared the average value trend of the dependent variables with the main explanatory variables among different regions. In this way, you can clearly identify trends of different variables in different regions. As for the unit of the variable, we also give explanations in the new line charts.
6. In addition to ‘Levin-Lin-Chiu’ method, we have added another two unit root test: ’Im, Pesaran and Shin’ and ‘ADF’. For your convenience, we have attached the original literature.
7. Yes. We have added views of similar papers in the literature review, conclusion and policy implications.
Above are the detailed corrections we made according to your comments point by point. Thanks again for your meticulous review and valuable suggestions to improve our manuscript. We hope that the revision has addressed all the issues in the old version. However, in the event that the modifications listed above DO NOT meet your expectations, we sincerely request you to give us another opportunity to revise the paper. We are very grateful for this opportunity given to us and would be more than happy to revise the paper until you are satisfied. We are looking forward to your positive response.
Best regards,
Yanchao Feng Xiaohong Wang* Wenchao Du Jun Liu
Reviewer 3 Report
This paper reports a statistical analysis of the associations between a number of variables that are surrogate measures for improvement in air quality and a number of variables that reflect measures of an ‘urban development quality’ (UDQ) metric. The various regressions and correlations are examined for 285 prefecture-level cities in China. The analysis includes examination of commonality or differences spatially across China.
I am an air pollution researcher, but the primary data analysis in this work is associated with economic variables, and a number of the statistical approaches and evaluations are beyond my specific expertise. I had not realised this when I agreed to take on review of this paper (apologies for that), so my review of this paper is from the perspective of a non-expert in this particular research field.
With this caveat of non-expertise, I nevertheless found this paper hard going to follow. A lot of data has clearly been gathered together and a lot of regressions and other statistical evaluations carried out. However, I am concerned that the paper reports much about significant or not-significant statistical associations without sufficient justification for the choices of the variables, nor adequate discussion of (a) issues of ‘multiple testing’ i.e. the likelihood that some significant results may not be genuine, or of (b) the underlying plausibility/causality of regression associations and trends.
The premise of the work is presumably that air pollution control is a causative factor for urban development quality (presumably in a positive sense) but I didn’t see a strong justification of why air pollution control is a putative causative factor for the chosen variables in the UDQ metric.
I also wonder whether the context of the research presented falls fully within the title of a journal that also includes “public health” in its title. Yes, air pollution control clearly impacts upon public health, but there is nothing directly in this paper that refers to public health.
Overall, and bearing in mind again my non-expert knowledge of this sort of work, I believe that the authors have relevant data and underlying scientific context to present, but I think this paper needs substantial work to improve it. The authors need to provide clearer focus to the scientific objectives of their study and to provide more incisive discussion of the extent to which their statistical findings reflect causality and the extent to which their findings can drive future national or regional policy directions. In addition, throughout the paper the quality of the written presentation needs to be edited to provide clear focus on key issues and points.
Some further comments (not comprehensive) are given below.
The abstract is not sufficiently informative. It refers to “urban development quality” but gives no indication of what is meant by this in the specific context of this study. Also, the final sentence of the abstract which states that “policy implications are presented” is not helpful to the reader – the abstract should provide the reader with at least a couple of the key ‘take home’ policy conclusions from their work.
The sentence in lines 34-35 beginning “Urban development quality is a comprehensive and dynamic concept” seems to be just jargon-ridden. What does it mean?
The paragraph in lines 81-89 is long-winded and not particularly specific to this work and can be edited for brevity.
The Introduction and the first part of the Methods section repeatedly uses the phrase “urban development quality” but without telling the reader what this actually means.
For the variables that refer to “removal rate” of SO2 or of smoke, what is the rate with respect to? I presume rate with respect to time, but what unit of time?
For the regression variables in Eq. 2, what is the rationale for choosing any of these particular economic variables? Were there other variables that could be chosen? Is there a reason for this particular total number of variables to use and not some other total number?
In a number of places the phrasings “is significant statistical” and “is not significant statistical” should be rephrased as “is statistically significant” and “is not statistically significant”.
In lines 225-226 there is repetition that data sources of relevant variables are presented in Table 1 and then that they are presented in Appendix B. Please clarify more specifically what is presented in Table 1 and what is presented in Appendix B.
Figures 1 to 5 should have scale bars in km not miles.
The phrasing “the government of smoke” should more likely mean “the regulation of smoke emissions”?
In Line 284 there is the phrasing “spatial feedback effects among different cities” which I don’t think makes any sense. This is a statistical association study, there are no feedbacks being investigated. Is it simply meant that there are spatial correlations?
In line 363, what is meant by “penal data”?
Author Response
Dear Reviewer,
We would like to thank you for your interest in our study and for the constructive comments. We have made the following revisions accordingly.
1. We admire you for your profound knowledge and the questions you raised. Although the variables in this paper come from a wide range of references, we still use unit root tests, correlation tests, model identification and other methods to ensure the rigor and objectivity of the analysis. The empirical results under different spatial weight matrix and spatial Durbin models also verify our conjecture.
2. There are many indicators of air pollution, but within the availability, the long-selling prefecture-level city data containing both production and removal is only sulfur dioxide and smoke and soot. We believe that with the increase of the available data, other air pollution control research (such as nitrogen dioxide) will gradually attract the attention of researchers.
3. The improvement of urban development quality is related to the development of public health. Although this paper dose not directly address public health, it still has practical value for public health. Thank you very much for your meaningful suggestions, and we will focus on the impact of air pollution control on public health in the subsequent studies.
4. According to your constructive comments, we reinterpret the connotation and dimension of urban development quality, and sort out the conclusions. On this basis, we summarize the policy recommendations to improve the pertinence of this article.
5. We have used line charts instead of maps and compared the average value trend of the dependent variables with the main explanatory variables among different regions. In this way, you can clearly identify trends of different variables in different regions. As for the unit of the variable, we also give explanations in the new line charts.
6. We have used a professional proofing service and have our revised manuscript checked by two native English speakers in the field of air pollution/statistics research to correct the grammar and format issues.
Above are the detailed corrections we made according to your comments point by point. Thanks again for your meticulous review and valuable suggestions to improve our manuscript. We hope that the revision has addressed all the issues in the old version. However, in the event that the modifications listed above DO NOT meet your expectations, we sincerely request you to give us another opportunity to revise the paper. We are very grateful for this opportunity given to us and would be more than happy to revise the paper until you are satisfied. We are looking forward to your positive response.
Best regards,
Yanchao Feng Xiaohong Wang* Wenchao Du Jun Liu
Round 2
Reviewer 1 Report
The manuscript has increased in quality but there are still issues with punctuation and grammar. I have tried to provide a “polishing” of the text identifying the ambiguous and unclear statements. Due to these shortcomings, I recommend minor revision because the paper cannot be published before making these corrections.
L10 delete “a” from a mixed regulations… have been
L40 delete coma before However and after government (L41)
L43 space after high (Same space issues for L50, L53, L55, L64, L74, L132, L138, L161, L197, L297. L379 …) – check throughout the manuscript
L59 Replace “there is relatively limited study”… with “there are few studies”
L72 space before Moreover
L77 I suggest to replace “inadequate” with “insufficient” or “scarce”.
L82 Replace comma after “types” with “such as”
L88 delete “so as”
L91 replace “is conductive” with “may lead”
L92 governments
L95 Space before This; This law represents well the impacts …
L97 what means agglomerate character? maybe it should be reformulated
L98 Replace So… with Consequently, we have selected Spatial Durbin…
L99 of both endogenous and exogenous interactions
L105-109 Reformulate in a clearer manner e.g., The spatial Durbin model (SDM) can examine the influence of the dependent variable affected by the local area variables, as well as the dependent and independent variables in neighboring areas, which is a general form of spatial lag model (SLM) and spatial error model (SEM). In this way, the spatial Durbin model (SDM) it is more suitable for the objectives of this study [30-32].
L121 Based on equation (1),… ; is as follows: (same for L147)
L163 as samples for this study.
L164 due to data unavailability
L165 other cities have data missing for certain years
L167 …2003-2016 has been used (figure 1).
L168 About Figure 1: For consistency, I suggest to use the same colors as in the graphs (figs. 2,3, 4 …) for each region. Either, you can modify the graphs using red for Eastern region, green for central and blue for Western or vice-versa.
L184 For supporting the reader, PCA should have more explanations and references regarding the related terms (see as example Statistical Analysis section from https://link.springer.com/article/10.1007/s11356-016-6734-x and related references)
L207 “in Figure 2. Figure 1 shows two main observations” I think it is about figure 2 not figure 1, and because a figure cannot show observations : “Figure 2 supports two main observations.”
L227 Figures
L228 check the figures numbers (first is 3-6 and then 2-5) because it is ambiguous.
L228 Three main observations can be drawn from Figures 2-5.
L229 and dust has shown…
L229 comma before while; same for L230 and L406
L248-L254 i.e.
L271 replace practice with methods
L273 indicate that…
L292 reveal
L330 delete repeating “of”
L393 Maybe should be useful to provide a synthetic table and associated comments that refine all the results by centralizing the main findings for each region and each key indicator. This will increase the value of the study and the understanding of major findings. Furthermore, it will support the conclusions and recommendations.
L400 (1) during …; (end with semicolon) (2) the impact of …; (3) the spatial … (end with point)
L402 “does” and again at L410
L403 delete “. were, .”
L416 “spatial dependences of the central region are” or “spatial dependence of the central region is”
L418 (1) to … same corrections as recommended at L400
L441 delete on point.
L443 In further research, an expansion …
L444 delete „so as”
L449 processes
L451 Any remaining errors are the responsibility of the authors.
L455 All of the authors have contributed to the writing and finalizing of the paper.
Author Response
Dear Reviewer, We would like to thank you for your interest in our study and for the constructive comments. We have made the following revisions accordingly. Thanks again for your meticulous review and valuable suggestions to improve our manuscript. We hope that the revised revision has addressed all the issues. We are looking forward to your positive response. Best regards, Yanchao Feng Xiaohong Wang* Wenchao Du Jun Liu

Reviewer 2 Report
This version is fine.
Author Response
Dear Reviewer, We would like to thank you for your interest in our study and for your recognition of our paper. As for the shortcomings of punctuation and grammar, we have taken a hard, critical review of this paper to correct the grammar and format issues. Thanks again for your meticulous review and valuable suggestions to improve our paper. Best regards, Yanchao Feng Xiaohong Wang* Wenchao Du Jun LiuReviewer 3 Report
This is a revised version of a submission that I recently reviewed.
In brief summary, the paper reports a statistical analysis of the associations between a number of variables that are surrogate measures for improvement in air quality and a number of variables that reflect measures of an ‘urban development quality’ (UDQ) metric. This is undertaken for 285 prefecture-level cities in China, grouped into three Chinese regions – ‘eastern’, ‘central’ and ‘western.’
As I indicated in my previous review, I am an air pollution researcher, but the nature of the data analysis in this work is predominantly with economic variables and using statistical methods with which I am not familiar.
With that caveat of non-expertise, I had raised a number of issues that I felt the reviewers needed to address. This included clearer explanations on why they had used the given number of specific variables that they had used, both the variables measuring aspects of air pollution control and the variables representing urban development, and why one would expect that air pollution control is a causative factor for the chosen variables in the UDQ metric. I also noted the lack of discussion on the issue of multiple testing or on the plausibility/causality of the significant associations described.
I also indicated that the whole paper needed an overhaul to ensure clearer focus and narrative on the points the authors wanted the reader to take away; and also for an improvement in the level of written English.
The revised paper has undergone extensive modification. The take-home implications are now much clearer. However, the cover letter that accompanies the revised submission does not directly respond to my comments on a point-by-point style and I am therefore unsure as to whether the authors have rejected (perhaps justifiably) some of my comments or simply ignored them. For example, there is no response to my comment on whether multiple testing is an issue given that there are tables and tables of correlation coefficients.
I also wonder whether some of the conclusions presented in the final section are really demonstrated beyond doubt within this work. The following is an example of a very powerful statement made in the conclusions: “However, if the alternative energy insufficient, unreasonably pursuing the reduction of sulfur dioxide will hinder the improvement of urban development quality.” Is the causative link underpinning this conclusion sufficiently robustly established?
The cover letter states that the revised paper has been read by two native English speakers and a professional proofing service. Unfortunately, the level of English still needs improvement – the first two sentences of the abstract both have grammatical errors, for example – but I acknowledge that English is not the first language of the authors. I spotted a number of other small errors as I read this through.
Overall, however, there is probably sufficient modification to consider publication, after the authors have been asked again to take a hard, critical review of their paper to ensure that they have full justification for their chosen data, and full justification of their conclusions from their results.
Author Response
Dear Reviewer, We would like to thank you again for your constructive comments. We admire you for your profound knowledge and the questions you raised. Thanks again for your meticulous review and valuable suggestions to improve our paper sincerely. We are looking forward to your positive response. Best regards, Yanchao Feng Xiaohong Wang* Wenchao Du Jun Liu
